# Cross-lingual transfer using phonological features for resource-scarce text-to-speech

*Johannes Abraham Louw*

Voice Computing Research Group,
Council for Scientific and Industrial Research,
Pretoria, South Africa

jalouw@csir.co.za

## Abstract

In this work, we explore the use of phonological features in cross-lingual transfer within resource-scarce settings. We modify the architecture of VITS to accept a phonological feature vector as input, instead of phonemes or characters. Subsequently, we train multispeaker base models using data from LibriTTS and then fine-tune them on single-speaker Afrikaans and isiXhosa datasets of varying sizes, representing the resource-scarce setting. We evaluate the synthetic speech both objectively and subjectively and compare it to models trained with the same data using the standard VITS architecture. In our experiments, the proposed system utilizing phonological features as input converges significantly faster and requires less data than the base system. We demonstrate that the model employing phonological features is capable of producing sounds in the target language that were unseen in the source language, even in languages with significant linguistic differences, and with only 5 minutes of data in the target language.

**Index Terms**: text-to-speech, resource-scarce, phonological features, cross-lingual

## 1. Introduction

Modern non-autoregressive text-to-speech (TTS) neural architectures, such as FastSpeech2 [1] and VITS [2], have made it possible to create natural-sounding TTS voices using significantly less data compared to the previously dominant concatenative synthesis or statistical parametric speech synthesis (SPSS) techniques [3]. However, these models still require a considerable amount of training data, which may not always be readily available in resource-scarce languages, the target language for building the voice. Generally, the availability of more data reduces the need for domain knowledge when constructing a TTS model, and vice versa. The term "resource-scarce" (or "low-resource") lacks a well-defined and universally agreed-upon definition. It encompasses various dimensions, including linguistic resources, acoustic resources, human capital, and computing resources. In this study, we adopt the term "resource-scarce" to refer to languages where a certain level of word segmentation, text normalization, and grapheme-to-phoneme (G2P) conversion functions are available, along with limited recordings of relatively high quality, typically 30 minutes or less.

Techniques that have emerged to address the challenge of data scarcity include self-supervised learning (SSL) and cross-lingual transfer. In self-supervised learning, a representation of speech is obtained, capturing low-level acoustic events, lexical knowledge, and syntactic and semantic information[4]. These representations can be further utilized to reconstruct the speech signal [5], making them suitable as an intermediate speech representation in a TTS model. This allows for the development of a two-stage TTS model. In the first stage, text is converted into an SSL representation, which is trained using relatively low-quality, but often abundant, automatic speech recognition (ASR) data. The second stage involves the conversion of the SSL representation into speech, and it can be trained using a smaller, high-quality TTS-style corpus. Recent examples of these models include [5], [6], [7] and [8]. These models show promise for building voices with limited data; however, to our knowledge, they have not yet been applied to cross-lingual voices.

In cross-lingual transfer, a TTS model is pre-trained using data from a resource-rich language, known as the "source language", to facilitate the learning of mappings between text and speech in resource-scarce languages, referred to as the "target language". This approach is grounded in the assumption that human languages share similarities in terms of pronunciation and semantic structures [9]. However, several challenges need to be addressed, including differences in phone inventories, representations, and language similarity. To handle these challenges, the differing phone inventories or input representations can be modeled using either *unified representations* or *separate representations*, as investigated in [10]. The findings of this study suggest that employing a unified input representation can enhance the accent and naturalness of multilingual TTS models. In this study, we compare a direct unified representation to phonological features as input in a cross-lingual TTS model. The question we ask is whether it is advantageous to use phonological features in cross-lingual transfer learning, particularly in resource-scarce scenarios. The key contributions of our work are as follows:

1. We modify a state-of-the-art TTS model to accept phonological features as input, deviating from the conventional character or phoneme-based schemes.

2. We train multiple TTS voices, simulating resource-scarce scenarios, and demonstrate that by utilizing phonological features, it is possible to train a cross-lingual model with as little as five minutes of data while achieving naturalness comparable to, or even better than, statistical parametric speech synthesis models.

## 2. Related Work

### 2.1. Phonological features

Phonological features (PFs) consist of a combination of articulatory and auditory/acoustic features [11], although the terms "phonological features" and "articulatory features" are sometimes used interchangeably [12]. According to [12], PFs can serve as a substitute for character models, where characters

are used to map phonemes or phones. Several recent studies [13, 14, 15, 16, 17] have focused on utilizing phonological features, either for cross-lingual phone inventory mapping or as direct input in TTS models. Phonological features are appealing as TTS input due to the potential for sharing features among phone identities and their universal nature.

Using phonological features in a TTS system offers advantages over phonetic models, including:

- *Simplified Data*: Phonological features provide a more abstract representation of speech sounds, reducing complexity by focusing on higher-level linguistic features. This simplification decreases computational requirements and training complexity.

- *Language Independence*: Phonological features are more language-independent, allowing for easier adaptation and transfer of TTS models to different languages with minimal adjustments, even when phonetic inventories and pronunciation rules differ.

- *Robustness to Variation*: Phonological features handle variations in speech sounds by capturing essential linguistic distinctions while allowing for flexibility. This enables the TTS system to accommodate diverse accents, dialects, and speaking styles, resulting in adaptable and natural-sounding synthesized speech.

- *Generalization and Out-of-Vocabulary Handling*: Phonological features improve generalization and out-of-vocabulary (OOV) handling within the TTS system. By focusing on abstract linguistic features, the model can extrapolate patterns and adapt to unseen or uncommon words, enhancing coverage and adaptability.

- *Reduced Data Requirements*: Phonological features may require less training data compared to full phonetic representations. Capturing fundamental linguistic characteristics, a smaller dataset can effectively train the TTS model, benefiting low-resource languages or situations with limited labeled phonetic data.

In linguistics, various schemes are used to model phonological features and analyze the distinct characteristics of speech sounds. These schemes include:

- *Binary Features*: Binary feature schemes categorize phonological features as present (+) or absent (-) for a speech sound. This approach simplifies representation by using binary distinctions for articulatory and other phonological features [18].

- *Multi-Valued Features*: Multi-valued feature schemes expand on binary features by allowing multiple values to represent phonological distinctions. These schemes incorporate additional values to capture nuanced phonetic properties [19].

- *Articulatory Features*: Articulatory feature schemes focus on the physical properties and movements of speech organs during speech production. They represent features based on articulatory gestures, including place and manner of articulation, voicing, and airstream mechanism.

- *Feature Geometry*: Feature geometry schemes depict phonological features as a hierarchical structure, capturing dependencies and relationships between features. This approach represents complex feature interactions and visualizes phonological processes and patterns [20].

In this work, we employ a phonological feature set based on the work in [16], with some modifications discussed in detail in Section 3. Our work is similar to the low-resource multilingual study conducted by [21], but we employ a different neural model, significantly less training data for the resource-rich source language, and we objectively and subjectively compare various sizes of resource-scarce datasets. [14] also utilized PFs for cross-lingual TTS, but their focus was on speaker adaptation, and they employed two orders of magnitude more training data. Additionally, [22] investigated PFs for cross-lingual TTS, albeit with a larger training dataset and an autoregressive model.

# 3. Proposed Method

The proposed model architecture is built upon the VITS model [2], which is an end-to-end TTS model operating in parallel. VITS utilizes a conditional variational autoencoder (VAE) to generate raw waveforms from text inputs. It incorporates normalizing flows [23] to handle the conditional prior distribution, and employs adversarial training [24] with a discriminator network to enhance the quality of synthesized speech. In this study, the standard VITS model has been modified in the following ways:

- **Phonological Features Encoder**: The original *Text Encoder* component of the VITS model has been replaced with a specialized *phonological features encoder*. This encoder is capable of processing a sequence of binary phonological feature vectors as input.

- **Decoder**: The original *Decoder* component of the VITS model has been substituted with a *Multi-Band inverse Short-Time Fourier Transform* (MB-iSTFT) decoder.

### 3.1. Models

#### 3.1.1. Baseline

The baseline model is the standard multi-speaker MB-iSTFT variant [1][25] of the VITS, using phones as input. The decision to adopt this model was influenced by several factors. Firstly, its foundation on the state-of-the-art VITS architecture ensures its alignment with cutting-edge methodologies in the field. Additionally, the model presents a favorable characteristic of being relatively lightweight, which contributes to computational efficiency. This advantage is further complemented by the notable speed of its inference, achieved through the implementation of an efficient MB-iSTFT decoder. Importantly, the model exhibits only a relatively minor degradation in naturalness compared to VITS, maintaining a satisfactory level of output quality. The model has a phoneme embedding lookup table with a dimension of 96.

#### 3.1.2. Proposed

The baseline architecture is enhanced through the substitution of the text encoder with a phonological encoder. The phonological encoder employs a sequential forward model with a non-linear activation, this enables the model to learn embeddings for multidimensional input features as opposed to the conventional one-dimensional character or phoneme-based input. The resulting output from the phonological encoder is then fed into the encoder of the VITS architecture in a similar manner to the text encoder in the baseline model. To ensure comparability with the baseline, a dimension of 96 is selected for this output.

---

[1]`https://github.com/MasayaKawamura/`
`MB-iSTFT-VITS`

Table 1: *Phonological features set used in this work.*

| Modifier | Category | Non-speech marker | Articulation | | Tongue position | Mouth | | | Voicing |
|---|---|---|---|---|---|---|---|---|---|
| | | | **Place** | **Manner** | | **Openness** | **Shape** | | |
| lengthened | consonant | silence | dental | plosive | back | close | rounded | | unvoiced |
| half-length | vowel | padding | postalveolar | nasal | central back | close close-mid | unrounded | | voiced |
| shortened | phoneme | question mark | velar | approximant | central | close-mid | | | |
| aspirated | | exclamation mark | palatal | trill | front central | mid | | | |
| ejective | | start of sentence | glottal | flap | front | open-mid | | | |
| | | end of sentence | uvular | fricative | | open-mid open | | | |
| | | syllable boundary | labiodental | lateral-approximant | | open | | | |
| | | word boundary | labial-velar | implosive | | | | | |
| | | phrase boundary | alveolar | vibrant | | | | | |
| | | | bilabial | click | | | | | |
| | | | alveolopalatal | lateral-fricative | | | | | |
| | | | retroflex | lateral-flap | | | | | |
| | | | pharyngal | | | | | | |
| | | | epiglottal | | | | | | |
| | | | labial-palatal | | | | | | |

### 3.2. Phonological Feature set

The phonological feature set utilized in this study is primarily based on the framework proposed by [16], with minor modifications. Our feature set comprises nine broad categories, eight of which directly correspond to the categories defined in the IPA. All features in the set are binary, resulting in a total of 60 combined features across all categories. Five of these features are considered "modifiers", which can either modify the preceding or succeeding phone feature vector. For instance, in the case of a lengthened vowel, both the "vowel" feature and the "lengthened" feature would be assigned a value of one.

In addition to the established categories, we have introduced an extra category called "non-speech marker". This category encompasses features that are either syntactic in nature or may influence the overall prosody of the investigated utterance. Table 1 gives the broad categories as well as the detailed features used in this work.

## 4. Experiment and Results

### 4.1. Data

The objective of this study was to develop a cross-lingual transfer model for TTS applications, specifically targeting resource-scarce languages. In this context, English was chosen as the resource-rich source language, while Afrikaans and isiXhosa were selected as the resource-scarce target languages. All the speech data includes the associated text transcriptions.

The language data employed consisted of a subset derived from the LibriTTS corpus[26]. Specifically, we focused on the male speakers' speech data since the resource-scarce datasets also consisted of male speakers. To reduce computational load during training, we filtered the utterances and selected only those containing 20 or fewer tokens. Randomly sampling from this filtered set, we obtained a total of 12 000 utterances, ensuring the manageability of the data within a single graphical processing unit (GPU).

For the resource-scarce data, we utilized subsets from the Lwazi corpora for Afrikaans and isiXhosa male speakers [27]. Similarly, we applied a filtering process, selecting only utterances with 20 or fewer tokens. To represent extreme resource scarcity, we chose three subsets for each language: 5 minutes, 15 minutes, and 30 minutes of speech data. These subsets were independently and randomly selected from each respective cor-

pus. Further details regarding the data used can be found in Table 2.

To construct the test and validation sets, we randomly selected 10 utterances from each language, ensuring they were not included in any of the training sets outlined in Table 2. Additionally, we chose 20 utterances from each language for the validation set.

Table 2: *Details of training datasets.*

| # | Lang | # Spk | Src | Dur | ID |
|---|---|---|---|---|---|
| 1 | Eng | 124 males | LibriTTS | 3:01:15.56 | libritts |
| 2 | Afr | 1 male | Lwazi | 0:05:00.15 | $afr_{5min}$ |
| 3 | | | | 0:15:05.28 | $afr_{15min}$ |
| 4 | | | | 0:30:00.74 | $afr_{30min}$ |
| 5 | | | | 3:33:26.53 | $afr_{all}$ |
| 6 | Xho | 1 male | Lwazi | 0:05:01.40 | $xho_{5min}$ |
| 7 | | | | 0:15:02.24 | $xho_{15min}$ |
| 8 | | | | 0:30:02.12 | $xho_{30min}$ |
| 9 | | | | 2:46:04.77 | $xho_{all}$ |

### 4.2. Pre-processing

The text annotations in the selected datasets (Table 2) underwent tokenization and normalization for each utterance. Linguistic descriptions were generated using the Speect TTS engine [28]. These descriptions consisted of phone strings with syntactic markup, including syllable breaks, word breaks, and phrase breaks. These syntactic markers align with the markers in the "non-speech marker"category in Table 1.

The baseline model took phone strings as input. To facilitate cross-lingual transfer with the baseline model, a *unified representation* for the phone inventory was established. This inventory encompassed all the distinct phones from the English, Afrikaans, and isiXhosa phone inventories. Instead of mapping between individual language inventories, the unified inventory represented a shared collection of unique phones that covered the complete phone inventory of the baseline models.

The proposed model utilized the phone strings with syntactic markup from the baseline model as input. These strings were then transformed into a phonological features vector using

the defined feature set in Tables 1. Each phone first maps to an IPA equivalent, which then is transformed into the phonological features vector.

Both the baseline and proposed models incorporated the practice of "interspersing" zero-valued vectors (or zeros for the baseline) into the input vector (or string for the baseline). This aided the alignment module (*monotonic alignment search*) of the VITS architecture.

All audio was down-sampled to 16 kHz at 16 bits per sample, and each utterance was normalized to the average power level of the dataset (as shown in Table 2).

Table 3: *Voices built for comparison between baseline and proposed models.*

| Voice # | Source | Target | | |
|---|---|---|---|---|
| | | **Model** | **Dataset** | **Identifier** |
| 1 | | | #2 | $afr_{5min\_ph}$ |
| 2 | | Baseline | #3 | $afr_{15min\_ph}$ |
| 3 | | | #4 | $afr_{30min\_ph}$ |
| 4 | #1 | | #5 | $afr_{all\_ph}$ |
| 5 | | | #2 | $afr_{5min\_pf}$ |
| 6 | | Proposed | #3 | $afr_{15min\_pf}$ |
| 7 | | | #4 | $afr_{30min\_pf}$ |
| 8 | | | #5 | $afr_{all\_pf}$ |
| 9 | None | HMM | #5 | $afr_{HMM}$ |
| 10 | None | Baseline | #5 | $afr_{all}$ |
| 11 | | | #6 | $xho_{5min\_ph}$ |
| 12 | | Baseline | #7 | $xho_{15min\_ph}$ |
| 13 | | | #8 | $xho_{30min\_ph}$ |
| 14 | #1 | | #9 | $xho_{all\_ph}$ |
| 15 | | | #6 | $xho_{5min\_pf}$ |
| 16 | | Proposed | #7 | $xho_{15min\_pf}$ |
| 17 | | | #8 | $xho_{30min\_pf}$ |
| 18 | | | #9 | $xho_{all\_pf}$ |
| 19 | None | HMM | #9 | $xho_{HMM}$ |
| 20 | None | Baseline | #9 | $xho_{all}$ |

## 4.3. Training

The performance of the proposed model was evaluated by comparing it to the base model. The training procedure involved building voices applying the two models (*baseline* and *proposed*) using the different datasets from Table 2. In each set, the first step was to train the *source language* model, representing the well-resourced language. Subsequently, the *target language* models were trained, representing the different variations of the resource-scarce language datasets. In total 20 voices were built from the combinations of the language datasets in Table 2 and the baseline and proposed models.

### 4.3.1. Source Language

The English data, representing the well-resourced language, was exclusively used to train the source language models. Adversarial training, where a discriminator $D$ that distinguishes between the output generated by the decoder $G$ and the ground truth waveform $y$, was used similar to [2] and [25].

For the baseline model, we set the number of sub-bands to 4. The up-sampling scale of the residual blocks was adjusted to match the resolution of each sub-band signal obtained through the analysis filters of the *Pseudo-Quadrature Mirror Filter* (PQMF) [29]. The *fast Fourier transform* (FFT) size, hop length, and window length of the iSTFT were kept consistent with the parameters used in [25]. Additionally, the parameters for calculating the STFT loss in the sub-bands were adopted from the same study.

Other training hyperparameters followed the approach used in [2], which involved employing the AdamW optimizer [30] with specific settings: $\beta_1 = 0.8$, $\beta_2 = 0.99$, and weight decay of $\lambda = 0.01$. The learning rate decay was scheduled to decrease by a factor of $\frac{0.9991}{8}$ in each epoch, starting with an initial learning rate of $2 \times 10^{-4}$. Mixed precision training was implemented on a single NVIDIA A6000 GPU. The batch size was set to 64, and the base model was trained for a total of 200k steps.

### 4.3.2. Target Language

The models for voices 1–8 and 11–18 in Table 3 were trained by initializing the weights with the source language voice trained in Section 4.3.1. Subsequently, training commenced with identical hyper-parameters as those used in the source language model. The training process involved fine-tuning the models for 100k steps, ensuring convergence while avoiding over-fitting.

For both Afrikaans and isiXhosa two extra voices were trained:

- We built a *hidden Markov model* (HMM)-based voice, utilizing all the language-specific data from Table 2 (datasets #5 and #9). This model served as the *anchor* for the subjective evaluation (see Section 4.4).

- Using the baseline model, we created voices (voice #10 and #20 in Table 3) using only the data available for the target language (datasets #5 and #9 from Table 2), the resource-scarce data. The purpose of building these voices was to establish a starting point for training voices *without* employing cross-lingual transfer, instead relying solely on the available data for the specific dataset. However, the training of these voice models did not converge, resulting in unintelligible and unusable synthesis in both subjective and objective tests. Furthermore, when we applied the proposed model to the same datasets using the same resource-scarce target datasets, the resulting voice qualities were the same, leading us to exclude these voices from further testing.

The test sets were also tokenized, normalised and converted into phoneme strings and PF vectors, after which they were synthesized for the objective and subjective evaluations. The reference and synthesized samples can be found here: `https://ghssw2023samples.github.io/experiment/`.

## 4.4. Evaluation

In order to quantitatively compare the proposed models with the baseline models we used the following objective measures:

- Mel-cepstral distortion (MCD), as defined in [31].
- $f_0$ Root mean squared error (RMSE), as defined in [32].
- $f_0$ Mean absolute error (MAE), this is similar to RMSE, but the MAE is less influenced by large outliers than RMSE.
- $f_0$ Voicing classification error (VCE), as defined in [32].
- Pearson correlation coefficient (PCC).

For subjective evaluation we conducted an online listening

test in the form of a web-based *MUltiple Stimuli with Hidden Reference and Anchor* (MUSHRA) [33] framework. The participants were asked to rate the *naturalness* in comparison to a reference sample on a scale from 0 (Very poor) to 100 (Completely natural).

The reference sample was a recording of human speech which was part of the test set, whilst the anchor was the HMM-based voice of Section 4.3.2. The participants were instructed to make sure that at least one of the rated samples was 100.

A total of 51 people participated in the perceptual evaluation, 27 for Afrikaans and 24 for isiXhosa. Only participants which were either first language speakers or fluent in the language were recruited. Post-screening was done to verify that participants scored the hidden reference stimuli 90 or more for more than $80\%$ of the tests and after post-screening there were 23 people for Afrikaans and 21 for isiXhosa.

The objective and subjective results are Tables 4 and 5 respectively. We also calculated the Wilcoxon rank-sum test to determine if the MUSHRA results are statistically significant between the different voice pairs. The *null hypothesis* for the Wilcoxon rank-sum test states that there is no difference between the distributions of the two samples, while the alternative hypothesis suggests a significant difference. The results are given in Table 6 for both Afrikaans and isiXhosa.

Table 4: *Objective results for both Afrikaans and isiXhosa: Mel-cepstral distortion (MCD), logarithmic scale, fundamental frequency ($f_0$) root mean squared error (RMSE), linear scale, fundamental frequency ($f_0$) mean absolute error (MAE), linear scale, fundamental frequency ($f_0$) voicing classification percentage error (VCE), and Pearson correlation coefficient of fundamental frequency ($f_0$) voicing classification (PCC).*

| Voice | MCD (dB) ↓ | $f_0$ | | | |
| | | RMSE ↓ (Hz) | MAE ↓ (Hz) | VCE ↓ (%) | PCC ↑ |
|---|---|---|---|---|---|
| $afr_{5min\_ph}$ | 6.86 | 28.94 | 18.42 | 15.62 | 0.401 |
| $afr_{5min\_pf}$ | 6.34 | 28.08 | 18.41 | 15.55 | 0.411 |
| $afr_{15min\_ph}$ | 6.38 | 28.14 | 17.35 | 15.71 | 0.423 |
| $afr_{15min\_pf}$ | 6.07 | 27.27 | 17.54 | 16.00 | 0.456 |
| $afr_{30min\_ph}$ | 6.24 | 27.61 | 17.49 | 13.45 | 0.498 |
| $afr_{30min\_pf}$ | 6.13 | 26.71 | 17.2 | 15.88 | 0.569 |
| $afr_{all\_ph}$ | 5.92 | 26.30 | 16.32 | 13.61 | 0.620 |
| $afr_{all\_pf}$ | 5.73 | 25.36 | 15.98 | 16.52 | 0.626 |
| $afr_{HMM}$ | 6.71 | 27.16 | 17.23 | 18.17 | 0.602 |

| Voice | MCD (dB) ↓ | $f_0$ | | | |
| | | RMSE ↓ (Hz) | MAE ↓ (Hz) | VCE ↓ (%) | PCC ↑ |
|---|---|---|---|---|---|
| $xho_{5min\_ph}$ | 6.45 | 12.45 | 7.65 | 11.55 | 0.601 |
| $xho_{5min\_pf}$ | 6.01 | 11.09 | 6.66 | 11.76 | 0.692 |
| $xho_{15min\_ph}$ | 6.04 | 10.53 | 6.16 | 10.93 | 0.760 |
| $xho_{15min\_pf}$ | 5.70 | 10.68 | 6.15 | 10.75 | 0.771 |
| $xho_{30min\_ph}$ | 5.80 | 10.18 | 5.92 | 11.26 | 0.798 |
| $xho_{30min\_pf}$ | 5.44 | 9.77 | 5.62 | 11.05 | 0.818 |
| $xho_{all\_ph}$ | 5.57 | 10.18 | 5.59 | 10.75 | 0.811 |
| $xho_{all\_pf}$ | 5.24 | 10.89 | 6.11 | 11.01 | 0.813 |
| $xho_{HMM}$ | 5.97 | 10.32 | 6.02 | 13.76 | 0.800 |

Table 5: *Mean MUSHRA subjective naturalness scores (0-100) for Afrikaans and isiXhosa with a 95% confidence level.*

| Voice | MUSHRA ↑ | Voice | MUSHRA ↑ |
|---|---|---|---|
| $afr_{5min\_ph}$ | $53.33 \pm 4.82$ | $xho_{5min\_ph}$ | $49.36 \pm 4.79$ |
| $afr_{5min\_pf}$ | $58.51 \pm 3.96$ | $xho_{5min\_pf}$ | $56.34 \pm 4.16$ |
| $afr_{15min\_ph}$ | $60.67 \pm 3.97$ | $xho_{15min\_ph}$ | $60.65 \pm 3.97$ |
| $afr_{15min\_pf}$ | $74.43 \pm 3.59$ | $xho_{15min\_pf}$ | $63.40 \pm 3.58$ |
| $afr_{30min\_ph}$ | $67.29 \pm 3.50$ | $xho_{30min\_ph}$ | $60.99 \pm 3.68$ |
| $afr_{30min\_pf}$ | $81.56 \pm 2.98$ | $xho_{30min\_pf}$ | $70.20 \pm 3.78$ |
| $afr_{all\_ph}$ | $76.04 \pm 3.42$ | $xho_{all\_ph}$ | $65.52 \pm 3.78$ |
| $afr_{all\_pf}$ | $86.27 \pm 2.36$ | $xho_{all\_pf}$ | $70.18 \pm 3.41$ |
| $afr_{HMM}$ | $51.30 \pm 3.27$ | $xho_{HMM}$ | $61.12 \pm 2.55$ |
| $afr_{Reference}$ | $96.95 \pm 0.91$ | $xho_{Reference}$ | $97.22 \pm 0.84$ |

## 5. Discussion and Conclusion

Analysis of Tables 4 and 5 reveals a consistent pattern for both Afrikaans and isiXhosa. The objective and subjective measures consistently indicate that the proposed model utilizing phonological features outperforms the baseline model.

As described in Section 4.3.2, a voice was trained for each variant and data subset of the target languages using the proposed model, excluding pre-training with the LibriTTS data (source language). However, due to significant issues of unintelligibility and noise artifacts, these voices and samples were excluded from the evaluations. This outcome was unexpected, especially for the Afrikaans and isiXhosa datasets that included all the data of those speakers. This finding further emphasizes the importance of pre-training on a resource-rich dataset, irrespective of the language.

Examining Table 6, we observe that there is no statistically significant difference in naturalness between an Afrikaans voice trained on `15 minutes` of data using phonological features as input (**G**) and a voice trained using all available data (`3.5 hours`) with phonetic features as input (**B**). Notably, an Afrikaans voice trained on `5 minutes` of data using the proposed phonological features as input (**I**) demonstrates statistical differentiation from an HMM-style voice (**A**), indicating superior performance based on the means of the MUSHRA test from Table 5.

Similar trends are observed for isiXhosa, despite the notable distinction between the source and target languages. In this case, a voice developed from `30 minutes` of speech using the proposed phonological features as input (**O**) is not found to be statistically different from a voice constructed from all available data (`2.75 hours`) using the baseline method (**L**).

Figure 1 illustrates the reference waveform and spectrogram of the recording containing the isiXhosa utterance *"Uya kubekwa kwindawo yokhuseleko lwamangqina yexeshana, ze emva koko, kuqwalaselwe isicelo sokhuseleko lwakho, isigxina."* This utterance can be loosely translated into English as *"You will be placed in temporary witness protection, and after that, your application for permanent protection will be considered."* Within the utterance, the isiXhosa word *"kuqwalaselwe"* (approximate English translation: *"considered"*) can be transcribed in IPA as [kʼuǃwalasɛlwɛ], where the third phone ([ǃ]) represents an alveolar click. The production of an alveolar click involves creating and suddenly releasing a vacuum between the tongue and the alveolar ridge, resulting in a distinctive click-

Table 6: *Statistical significant differences for Afrikaans and isiXhosa between the different voice pairs ($p < 0.05$). Pairs that are statistically significantly different are donated with a diamond marker (◇) while pairs that are not statistically significantly different are donated with a filled diamond marker (◆).*

| | A | B | C | D | E | F | G | H | I | J |
|---|---|---|---|---|---|---|---|---|---|---|
| **A** | · | ◇ | ◇ | ◇ | ◇ | ◇ | ◇ | ◆ | ◇ | ◇ |
| **B** | ◇ | · | ◇ | ◇ | ◇ | ◇ | ◆ | ◇ | ◇ | ◇ |
| **C** | ◇ | ◇ | · | ◆ | ◆ | ◇ | ◇ | ◇ | ◇ | ◇ |
| **D** | ◇ | ◇ | ◆ | · | ◇ | ◇ | ◇ | ◇ | ◇ | ◇ |
| **E** | ◇ | ◇ | ◆ | ◇ | · | ◇ | ◇ | ◇ | ◇ | ◇ |
| **F** | ◇ | ◇ | ◇ | ◇ | ◇ | · | ◇ | ◆ | ◆ | ◇ |
| **G** | ◇ | ◆ | ◇ | ◇ | ◇ | ◇ | · | ◇ | ◇ | ◇ |
| **H** | ◆ | ◇ | ◇ | ◇ | ◆ | ◇ | ◇ | · | ◆ | ◇ |
| **I** | ◇ | ◇ | ◇ | ◇ | ◇ | ◆ | ◇ | ◆ | · | ◇ |
| **J** | ◇ | ◇ | ◇ | ◇ | ◇ | ◇ | ◇ | ◇ | ◇ | · |

| | | | | | | |
|---|---|---|---|---|---|---|
| **A** | $afr_{HMM}$ | **D** | $afr_{30min\_ph}$ | **G** | $afr_{15min\_pf}$ |
| **B** | $afr_{all\_ph}$ | **E** | $afr_{30min\_pf}$ | **H** | $afr_{5min\_ph}$ |
| **C** | $afr_{all\_pf}$ | **F** | $afr_{15min\_ph}$ | **I** | $afr_{5min\_pf}$ |
| **J** | $afr_{Reference}$ | | | | |

| | K | L | M | N | O | P | Q | R | S | T |
|---|---|---|---|---|---|---|---|---|---|---|
| **K** | · | ◆ | ◇ | ◆ | ◇ | ◆ | ◆ | ◇ | ◆ | ◇ |
| **L** | ◆ | · | ◆ | ◆ | ◆ | ◆ | ◆ | ◇ | ◇ | ◇ |
| **M** | ◇ | ◆ | · | ◇ | ◆ | ◇ | ◆ | ◇ | ◇ | ◇ |
| **N** | ◆ | ◆ | ◇ | · | ◇ | ◇ | ◆ | ◇ | ◆ | ◇ |
| **O** | ◇ | ◆ | ◆ | ◇ | · | ◇ | ◇ | ◇ | ◇ | ◇ |
| **P** | ◆ | ◆ | ◇ | ◇ | ◇ | · | ◆ | ◇ | ◆ | ◇ |
| **Q** | ◆ | ◆ | ◆ | ◆ | ◇ | ◆ | · | ◇ | ◇ | ◇ |
| **R** | ◇ | ◇ | ◇ | ◇ | ◇ | ◇ | ◇ | · | ◆ | ◇ |
| **S** | ◆ | ◇ | ◇ | ◆ | ◇ | ◆ | ◇ | ◆ | · | ◇ |
| **T** | ◇ | ◇ | ◇ | ◇ | ◇ | ◇ | ◇ | ◇ | ◇ | · |

| | | | | | | |
|---|---|---|---|---|---|---|
| **K** | $xho_{HMM}$ | **N** | $xho_{30min\_ph}$ | **Q** | $xho_{15min\_pf}$ |
| **L** | $xho_{all\_ph}$ | **O** | $xho_{30min\_pf}$ | **R** | $xho_{5min\_ph}$ |
| **M** | $xho_{all\_pf}$ | **P** | $xho_{15min\_ph}$ | **S** | $xho_{5min\_pf}$ |
| **T** | $xho_{Reference}$ | | | | |

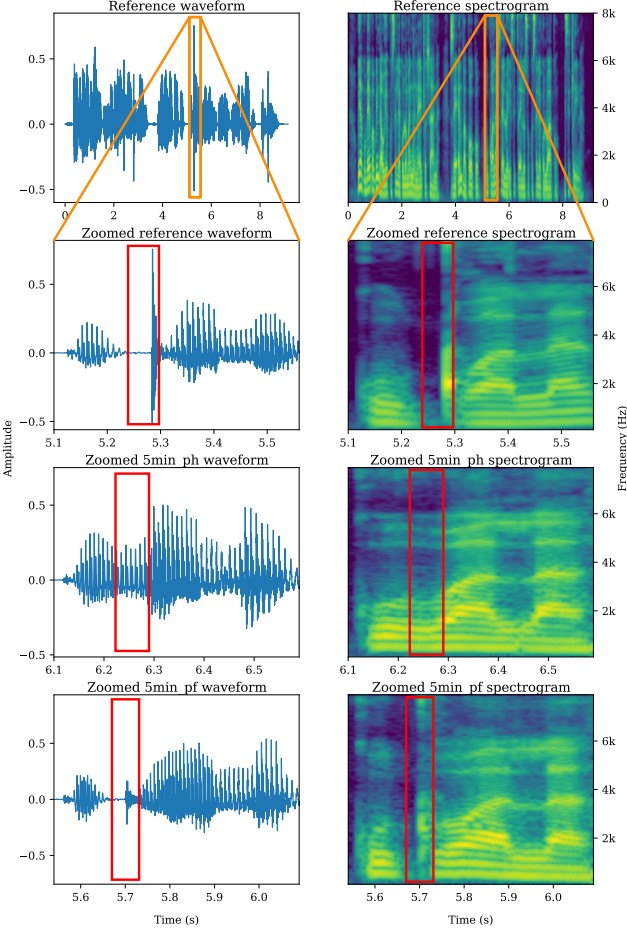

Figure 1: *Example of click ([!]) with 5 minutes of data.*

ing sound. This phonetic feature poses a challenge in cross-lingual transfer due to differences in phone inventories across languages, with no equivalent sound in English.

The second row of plots in Figure 1 provides a closer examination of the click ([!]) by presenting a zoomed-in section of the reference waveform and spectrogram. The third row of plots showcases the realization of the click in a synthesized sample generated by the voice built using the `5 minute` dataset and the baseline model (which utilizes a shared phone inventory, as described in Section 4.2). Similarly, the fourth row of plots displays the click from a synthesized sample produced by the voice built using the `5 minute` dataset and the proposed model.

Both the waveform and spectrogram clearly depict the silent section preceding the abrupt release of the click and the sharp boundary where the click begins, resembling the perceptual qualities of the click in the reference recording. This finding is significant as it demonstrates the ability of the proposed model to learn a phonetic feature with limited examples in the target language (isiXhosa) and no examples in the source language (English). The test set utterances consisted of a total of 1423 phones, with the isiXhosa phone inventory encompassing 45 distinct phones (excluding non-speech markers, as indicated in Table 1). Out of these 1423 phones, only 5 instances of the

click ([!]) were present in the test set, which the proposed model effectively learned, accounting for a mere $0.35\%$ of the total number of phones in the test set.

While this study has demonstrated the improvement in naturalness of synthesized speech through the utilization of phonological features in cross-lingual TTS for resource-scarce languages, there are several avenues for future research that can further advance the field and address remaining challenges. These avenues include exploring alternative transfer learning techniques such as multi-lingual transfer learning, unsupervised transfer learning, or domain adaptation methods to enhance the generalization and adaptation capabilities of TTS systems. Additionally, conducting robustness analysis to assess the performance of the proposed model under diverse conditions and potential biases is crucial.

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
