# OpenReview forum: "Cross-lingual transfer using phonological features for resource-scarce text-to-speech"
_Interspeech.org/2023/Workshop/SSW — SSW12_

### Official Review · Reviewer_g4Jb · 2023-05-30
**Phonological features as input to VITS**

**Rating:** 6
**Confidence:** 5

**Review:**

- Summary: Use phonological features as input to VITS-based TTS model then evaluated its performance (fine-tuning a base model trained from an English dataset to low-resource languages) in objective and subjective metrics.

- Quality: Well written.

- Clarity: Good.

- Originality: Minor.  Very similar to the previous work [13,25,26].  Main differences are the eval metrics, different types of models.

- Pros: More evals and discussions.

- Cons: Minor originality.  Cannot find difference other than base models.

---

### Official Review · Reviewer_N7j2 · 2023-05-31
**Clearly written and well organized paper with convincing results**

**Rating:** 7
**Confidence:** 4

**Review:**

This paper describes a method to synthesize speech in data-scarce conditions, meaning that there is little acoustic data for the target language. The authors propose to pre-train a VITS model using a large amount of English data using either 1) phones or 2) phonological features. Then they fine-tune the models with the target languages, Afrikaans and isiXhosa (separately) using different amounts of data. The objective and subjective quality if the synthesized speech is assessed. The results show that the systems using the phonological features outperform the systems using phonemes, and also the baseline HMM-based TTS system trained using all the target data.

The paper is clearly written and well organized. Although the idea of using phonological features is not new, the results are convincing in providing good quality with very little data. Overall, this work is useful in demonstrating how to build a TTS system in data-scarce conditions.

Detailed comments:

Sec. 1: “using an order of magnitude less data than the previous commercially dominant concatenative synthesis or statistical parametric speech synthesis (SPSS) techniques [3].” This might not be completely true, especially in the case of HMM-based speech synthesis where adaptation with very small amounts of data was prevalent.

Sec. 5: “and it’s influence” —> and its influence

---

### Decision · Program_Chairs · 2023-06-14

**Decision:**

Accept

**Comment:**

SSW2003 received 45 papers. The acceptance rate is 82%. We are pleased to inform you that your paper has been accepted by the SSW2023 Program Committee. Please read the reviews carefully and submit your camera-ready paper by June 28th. Most reviewers performed a detailed review. Please answer to their questions and consider their comments. Note that camera-ready papers are credited with one extra page to allow authors to consider reviewers’ suggestions. So max 7 pages in total including figures & refs.
The deadline for submitting the revised version (with full non-anonymized authors and refs!) is 28th June.